# BGN/TLR4/NF-κB Mediates Epigenetic Silencing of Immunosuppressive Siglec Ligands in Colon Cancer Cells

**DOI:** 10.3390/cells9020397

**Published:** 2020-02-09

**Authors:** Hsiang-Chi Huang, Bi-He Cai, Ching-Shu Suen, Hsueh-Yi Lee, Ming-Jing Hwang, Fu-Tong Liu, Reiji Kannagi

**Affiliations:** 1Institute of Biomedical Sciences, Academia Sinica, Taipei 11529, Taiwanbigbiha@ibms.sinica.edu.tw (B.-H.C.); csuen@gate.sinica.edu.tw (C.-S.S.); sakon@ibms.sinica.edu.tw (H.-Y.L.); mjhwang@ibms.sinica.edu.tw (M.-J.H.); ftliu@ibms.sinica.edu.tw (F.-T.L.); 2Department of Medicine, College of Medicine, I-Shou University, Kaohsiung 82445, Taiwan

**Keywords:** TLR, NF-κB, sialyl 6-sulfo Lewis^x^, disialyl Lewis^a^, BGN, Siglec-7, SLC26A2, ST6GalNAc6

## Abstract

Human Toll-like receptor (TLR) signaling plays a vital role in intestinal inflammation by activating the NF-κB pathway. By querying GENT2 datasets, we identified the gene expression level of TLR2 and TLR4 as being substantially increased in colorectal cancer. Introduction of shRNAs for TLR4 but not TLR2 dramatically recovered disialyl Lewis^a^ and sialyl 6-sulfo Lewis^x^ glycans, which are preferentially expressed in non-malignant colonic epithelial cells and could serve as ligands for the immunosuppressive molecule Siglec-7. We screened several TLR4 ligands and found that among them BGN is highly expressed in cancers and is involved in the epigenetic silencing of Siglec-7 ligands. Suppression of BGN expression substantially downregulated NF-κB activity and the marker H3K27me3 in the promoter regions of the SLC26A2 and ST6GalNAc6 genes, which are involved in the synthesis of those glycans, and restored expression of normal glycans as well as Siglec-7 binding activities. We show that in the presence of TLR4, inflammatory stimuli initiate a positive loop involving NF-κB that activates BGN and further enhances TLR4 activity. Present findings indicate a putative mechanism for the promotion of carcinogenesis by loss of immunosuppressive ligands by the BGN/TLR4/ NF-κB pathway.

## 1. Introduction

Cancer is the second leading cause of mortality worldwide and was accountable for approximately 9.6 million deaths in 2018. Colorectal cancer is the third-leading cause of cancer mortality in both men and women [1]. Inflammation is a critical risk factor for human colorectal cancer initiation and progression, and most tumor-promoting cytokines are activated via the NF-κB pathway. Aberrant NF-κB activation was detected in more than 50 percent of colorectal tumors [2]. All Toll-like receptors (TLRs) except TLR3 activate NF-κB signaling through the MyD88 pathway [3]. MyD88/TLR signaling is overexpressed and associated with intestinal tumor formation and progression [4].

Aberrant glycosylation occurs in all sorts of human cancers [5]. Previously, we demonstrated that healthy colonic epithelial cells express the immunosuppressive Siglec-7 ligands disialyl Lewis^a^ and sialyl 6-sulfo Lewis^x^, and we showed that the silencing of SLC26A2 and ST6GalNAc6, which are involved in their synthesis, upon malignant transformation leads to the appearance of two cancer-associated glycans called sialyl Lewis^a^ and sialyl Lewis^x^ [6,7,8]. Siglec-7 is a family of immunosuppressive molecules present in colonic mucosal dendritic cells and macrophages and previously we proposed that they prevent excess activation of mucosal immune cells and maintain immune homeostasis of the colonic mucosal membranes by suppressing the production of pro-cancer inflammatory molecules such as COX2 [9]. We investigated the mechanisms involved in the epigenetic silencing of disialyl Lewis^a^ and sialyl 6-sulfo Lewis^x^ in human colon cancers. Enhancer of zeste (EZH2) is an H3K27 methyltransferase that is highly elevated in colorectal cancer [10] and mediates H3K27me3, a type of histone modification, that serves as an epigenetic marker for SLC26A2 or ST6GalNAc6 gene silencing [11]. The NF-κB (p65)/EZH2 axis plays a vital role in epigenetic silencing of the synthesis of immunosuppressive Siglec ligand glycans [12].

TLR signaling plays a crucial role in intestinal inflammation, but the role of TLRs in the regulation of immunosuppressive ligand expression remains unknown. Among all the TLRs, expression of TLR2 and TLR4 mRNA is increased significantly in colorectal cancer based on the bioinformatics analysis in the GENT2 database. In this study we found that the introduction of shRNAs for TLR4 but not TLR2 significantly recovers Siglec ligands. Furthermore, we also propose that BGN (biglycan) is one of the potential TLR4 ligands that are highly elevated and involved in the epigenetic silencing of Siglec ligands in colorectal cancer.

## 2. Materials and Methods

### 2.1. Cell Cultures

Human colon cancer cell lines such as DLD-1, HT-29, and LS174T (ATCC, Manassas, VA, USA) were cultured and maintained in RPMI1640 or DMEM medium (Invitrogen, Carlsbad, CA, USA) supplemented with 10% FBS (Invitrogen). Bay11-7085 (NF-κB inhibitor), TPCA-1 (NF-κB inhibitor), and TAK-242 (TLR4 inhibitor) were purchased from Selleckchem (Houston, TX, USA).

### 2.2. Clinical Samples, Real-Time Quantitative PCR (RT-qPCR)

Tumor specimens for RT-PCR analysis were obtained from 10 patients with primary colorectal cancer at surgical operation and processed as described previously [8]. These samples were collected in accordance with the guidelines of the World Medical Association’s Declaration of Helsinki. Total RNA was extracted using the Quick-RNA MiniPrep kit from ZYMO Research (Irvine, CA, USA). The ToolsQuant II Fast RT Kit (TOOLS, Taiwan) was used to reverse transcribe total RNA into cDNA. RT-qPCR was conducted in a CFX Connect system (Bio-Rad, Hercules, CA, USA) using the Evagreen master mix kit (Bio-Rad) with a cDNA template. Primers used for RT-qPCR are listed in Appendix A. GAPDH was applied as an internal control.

### 2.3. Immunoprecipitation and DNA Affinity Precipitation Assay (DAPA)

HT-29 and LS174T cells were cultured and maintained in DMEM medium supplemented with 10% FBS. After 48 hours, the culture medium was collected and concentrated ~ 100-fold using Amicon Ultra-15, 30kDa (UFC903008; Millipore, Billerica, MA, USA), and immunoprecipitated (IP) with anti-BGN antibody (ab94460; Abcam, Cambridge, MA, USA). The DAPA experiments were performed as previously described [12]. Antibodies specific for Western blot included EZH2 (21800-1-AP; Proteintech), p65 (39369; Active Motif), p-p65 (3033S; Cell Signaling) and BGN antibody (ab94460; Abcam). 

### 2.4. Bioinformatics Analysis of Gene Expression

The gene expression of TLR1-TLR10, BGN, HSPD1, HMGB1, S100A8, S100A9, TNC, SDC2 (HSPG1), HSPG2, HYAL1, HYAL2, HAS1, HAS2, and HAS3 in normal (*n* = 397) and colon cancer samples (*n* = 3775) was analyzed through the Gene Expression database of Normal and Tumor tissues (GENT2). Gene expression data was downloaded from the GEO public repository using the U133Plus2 (GPL570) platform [13]. The amounts of TLR1-TLR10 and TLR4 ligands in the colorectal cancer samples were compared to those of normal colon samples by Gene Expression Profiling Interactive Analysis (GEPIA2) [14]. Gene expression datasets were downloaded from large-scale RNA sequencing resources at the Broad Institute Cancer Genome Atlas (TCGA) GDAC Firehose as previously described [12,15]. The RNA expression data for healthy and cancer tissues as well as the expression data for BGN, HSPD1, DFEB1, HMGB1, HAS1, HAS2, HAS3, HYAL1, HYAL2, SDC2 (HSPG1), HSPG2, S100A8, S100A9, SFTPA1, and TNC were downloaded from TCGA datasets.

### 2.5. Chromatin Immunoprecipitation (ChIP)

The ChIP experiments were conducted as previously described [12]. Antibodies from Active Motif (Carlsbad, CA, USA) specific for ChIP experiments included p65 (39369) and H3K27me3 (39155). Primers used to detect the EZH2, SUZ12, EED, SLC26A2, ST6GalNAc6, and BGN regulatory regions are listed in Appendix A.

### 2.6. Administration of shRNA for BGN, p65, TLR2, and TLR4

The pLKO.1-based lentivirus for p65 (RELA) (TRCN0000014684), BGN-1 (TRCN0000152593), BGN-2 (TRCN0000156631), TLR2-1 (TRCN0000057019), TLR2-2 TRCN0000057021), TLR4-1 (TRCN0000056894), and TLR4-2 (TRCN0000056895) were obtained from the National RNAi Core Facility Platform at the IMB/GRC, Academia Sinica. The cells were infected with lentivirus (MOI=3) with 8 g/ml polybrene. After 1 d infection, the medium was changed to fresh medium and cultured for an additional 1 d. Selection were performed by culturing the cells with 5 g/ml puromycin for 7 d. The cells expressing scrambled shRNA (pLAS.Void) were applied as a control.

### 2.7. Flow Cytometry and Recombinant Siglec Binding Assays

Glycan-specific monoclonal antibodies were used to detect the expression of glycans on cell surface by flow cytometry analysis. The antibody FH7 (murine IgG3) [16] was applied for assessment of disialyl Lewis^a^ expression. The antibody G72 (murine IgM) was prepared as previously described [17] and was employed for assessment of sialyl 6-sulfo Lewis^x^ levels. Binding of recombinant Siglecs to LS174T was ascertained by flow cytometric analyses as described [12,18].

### 2.8. Reporter Constructs and Luciferase Assays

The human BGN promoter (−823/−1) was amplified by PCR. Genomic DNA was extracted from LS174T cells using the Genomic DNA purification kit (Molecular Research Center, Cincinnati, OH, USA) and served as a template. The deleted p65 binding motif BGN construct (BGN p65 del.) had a deletion at bases −647 to −639 in the BGN promoter. The amplicon was purified with the GeneJET Plasmid Miniprep kit (Thermo Fisher Scientific, Vilnius, Lithuania) and cloned into the pGL3-basic vector (Promega, Madison, WI, USA). We synthesized EZH2, SUZ12, and EED regulatory regions and constructed those fragments into a pGL3 vector system. The mutated p65 binding motif EZH2 construct (EZH2 p65m) had a mutation at bases +994 to +841 (from CCCCTAAAGC to TTTTTTTTTT) in the EZH2 intron 1 p(+885/+876). The mutated p65 binding motif SUZ12 construct (SUZ12 p65m) had a mutation at bases −114 to −105 (from GGGGAATCCGC to AAAAAATAAAA) in the SUZ12 promoter p(−231/+69) and a mutation at bases −219 to −210 (from GGGTACTTTCC to AATACAAAAA) in the EED promoter p(−348/−49). pBabe-Puro-IKBalpha-mut (Addgene plasmid #15291) [19] is a dominant-negative mutant NF-κB inhibitor (IkBm) and inhibits NF-κB activity; IkBm was cloned into pcDNA3.1. Sequencing (Applied Biosystems 3730XL system) was performed to confirm the sequence of each cloned regulatory region. To assess the promoter activity, cells were co-transfected with the pRL-TK vector (Promega) and the BGN or BGN p65 del. promoter construct vector. For the luciferase assay, X-tremeGENE HP DNA Transfection Reagent (Roche) was used to co-transfect colon cancer cells with an expression vector (pcDNA3.1) encoding IkBm and a vector with the BGN promoter construct. After 1 d, luciferase activity was evaluated by using the Dual-Luciferase Reporter Assay System kit (Promega).

### 2.9. Statistical Analysis

Prism 5 software (Prism Windows 5.00, GraphPad, La Jolla, CA, USA) was used to assess the statistical differences between two groups with a *t*-test (two-tailed). All results are presented as the mean ±SD, and *p* < 0.05 was considered statistically significant (*, *p* < 0.05; **, *p* < 0.01; ***, *p* < 0.001).

## 3. Results

### 3.1. TLR4 Silences Immunosuppressive Siglec Ligand Expression

To determine which human TLRs may play roles in the suppression of Siglec ligand synthesis, we first analyzed the amounts of human TLR1-TLR10 in colorectal cancer tissues in the GEPIA2 database (http://gepia2.cancer-pku.cn/#index). TLR4 turned out to be the dominant TLR in colorectal cancer tissues (Figure 1A). Next, we looked at cancer-associated alteration of gene expression levels in all TLRs in the GENT2 database (http://gent2.appex.kr/gent2/), and only TLR2 and TLR4 were increased significantly in colon cancer samples (Figure 1B). TLR4 mRNA tended to increase in human colon cancer cells compared to non-malignant colonic epithelial cells prepared from the same patients (Figure 1C). To identify whether TLR2 or TLR4 is responsible for the silencing of immunosuppressive Siglec ligands, we analyzed changes in disialyl Lewis^a^ or sialyl 6-sulfo Lewis^x^ glycan expression by knocking down TLR2 or TLR4 by applying specific shRNA for TLR2 or TLR4 in colon cancer cells. DLD-1 and LS174T cells having high expression levels of TLR2 compared to HIEC6 cells (healthy intestinal epithelial cells) were selected for the TLR2 knocking-down experiments. Similarly, HT-29 and LS174T having high expression levels of TLR4 were selected for the TLR4 knocking-down experiments (Figure 1D). Administration of shRNAs for TLR2 (Figure 1E) or TLR4 (Figure 1F) dramatically decreased the mRNA levels of TLR2/4, and suppression of TLR4 but not TLR2 significantly induced SLC26A2 (Figure 1G) and ST6GalNAc6 mRNA levels (Figure 1H). Moreover, silencing of TLR4 but not TLR2 markedly recovered disialyl Lewis^a^ or sialyl 6-sulfo Lewis^x^ expression on the cancer cell surface (Figure 1I–L). 

Taking these findings together, TLR4, which is the most dominant form of TLRs and increased significantly in colon cancer, could serve as a crucial factor for suppression of disialyl Lewis^a^ or sialyl 6- sulfo Lewis^x^ in colon cancer cells.

### 3.2. Identification of BGN as a Potential TLR4 Ligand Involved in Suppression of Siglec Ligand in the Early Stage of Colon Cancer Carcinogenesis

Several endogenous TLR4 ligands have been proposed, including BGN, HSPD1, HMGB1, TNC, S100A8, S100A9, heparan sulfate, and hyaluronan [20]. To investigate which TLR4 ligand is potentially involved in the activation of the TLR4 pathway in colorectal cancers, we retrieved BGN, HSPD1, HMGB1, S100A8, S100A9, TNC, hyaluronidase (HYAL1 and HYAL2), heparan sulfate proteoglycan (SDC2 also called HSPG1, and HSPG2), hyaluronan synthase (HAS1, HAS2, and HAS3), and SFTPA1 gene expression data in healthy colon control and colon cancer from the GENT2 database. We identified that BGN, S100A8, S100A9, and TNC expression levels were elevated significantly in colon cancer samples (Figure 2A). Among those candidates, BGN is the most dominant form in COAD and READ (Figure 2B). BGN expression tends to increase in different cancers, and colon cancer expresses most (Appendix A). In addition, we also confirmed that BGN mRNA was elevated most significantly in colorectal cancer compared to healthy control in the TCGA database (Appendix A). There was a dramatic increase in mRNA levels of BGN in stage I cancer samples, and this was extremely statistically significant *p* < 0.001 (Figure 2C). 

This increase in BGN mRNA showed a close chronological correspondence with the sharp decrease in ST6GalNAc6 and SLC26A2 mRNAs between healthy colon and stage 1 samples (Appendix A). Overall survival analysis demonstrated that BGN was significantly associated with the worse overall survival of colon cancer (Figure 2D). Next, we investigated whether suppression of BGN can restore the levels of the normal glycan disialyl Lewis^a^ and sialyl 6- sulfo Lewis^x^ on the cell surface. Administration of shRNAs for BGN downregulated mRNA of BGN (Figure 2E) in colon cancer cells. Additionally, the administration of shRNAs for BGN led to the expression of normal glycans on colon cancer cells (Figure 2F). Moreover, we investigated Siglec-7 binding activity by the administration of shRNA for BGN. Consistently, the induction of shBGN significantly restored Siglec-7 binding in the colon cancer cells (Figure 2G). 

Taking these findings together, it is suggested that BGN is the dominant form of TLR4 ligands and involved in colorectal carcinogenesis. The introduction of shRNA for BGN could restore immunosuppressive Siglec-7 ligand and its binding activity in colon cancer cells.

### 3.3. Suppression of BGN Inactivates p65 and Decreases PRC2 Activity in Promoter Regions of SLC26A2 and ST6GalNAc6

We previously showed that the NF-κB subunit (p65) plays a vital role in the epigenetic silencing of Siglec-7 ligands by triggering EZH2-containing PRC2 complex to promoter regions of SLC26A2 and ST6GalNAc6 [12]. BGN is a small leucine rich proteoglycan ECM protein. Soluble BGN directly interacts with TLR-2 and TLR-4 [21]. Mechanistically, BGN is known to rapidly activate NF-κB by engaging TLR2 and TLR4 [22]. In HT-29, suppression of BGN can decrease NF-kB activity [23]. To clarify if BGN is secreted to cultured medium and serves as a potential ligand for TLR2/4 in colon cancer cells, we looked at BGN in HT-29 and LS174T cultured medium using the immunoprecipitation–Western blotting method (Figure 3A). Here we showed that the introduction of shRNAs for BGN significantly decreases activated p65 (p-p65) protein level (Figure 3B). Constitutive NF-κB activation was observed in 67% of colorectal cancer cell lines [24], and we also confirmed that some colon cancer cells such as DLD-1, HT-29, and LS174T display much higher p65 activity compared to normal intestinal cells (HIEC6) based on reporter assays (Appendix A). EZH2 is active as a methyltransferase only in the form of the PRC2 complex [25]. Indeed, we showed p65 could bind to regulatory regions of PRC2 elements such as EZH2, SUZ12, and EED by the ChIP/RT-qPCR results (Figure 3C). ChIP-seq data from the UCSC (The University of California, Santa Cruz) database also suggested that p65 can bind to regulatory regions of PRC2 components (Appendix A). Mutation in the putative p65 binding sites in the p(+994/+841) construct for EZH2, p(−231/+69) construct for SUZ12, and p(−348/−49) construct for EED significantly decreased reporter activity (Figure 3D). Administration of Bay11-7085, an NF-κB inhibitor, also downregulated EZH2, SUZ12, and EED mRNA levels (Figure 3E). Moreover, the introduction of shRNAs for BGN substantially reduced H3K27me3 levels in the promoter regions of both SLC26A2 and ST6GalNAc6 based on ChIP/RT-qPCR results (Figure 3F). Taking these findings together, BGN may serve as a trigger to initiate NF-κB/PRC2-mediated epigenetic silencing of SLC26A2 and ST6GalNAc6.

### 3.4. Roles of TLR4 in Mediating NF-κB/BGN Loop for Epigenetic Silencing of SLC26A2 and ST6GalNAc6

Previously, we showed that the introduction of shRNAs for BGN decreased NF-κB activity and restored immunosuppressive Siglec ligand expression by decreasing PRC2 activity in cultured colon cancer cells. NF-κB subunit (p65) also regulates BGN expression [26]. To examine whether TLR4 could regulate BGN through NF-κB activity, we used a small molecule inhibitor, TAK-242, that binds selectively to TLR4 [27]. Administration of TAK-242 downregulated NF-κB and BGN promoter activities (Figure 4A,B). Consistently, the induction of shRNAs for TLR4 dramatically decreased BGN protein levels (Figure 4C). To further confirm involvement of NF-κB subunit (p65) in BGN activity, we performed a ChIP assay to verify that p65 can bind to the BGN promoter (Figure 4D). In vitro DNA affinity pull-down (DAPA) assay verified the p65 binding region in BGN promoter (Figure 4E). Furthermore, the deletion of the p65 binding motif led to lower BGN promoter activity (Figure 4F). Treatments using NF-κB inhibitors, Bay11-7085 and TPCA-1, also confirmed that NF-κB regulates BGN mRNA (Figure 4G) and protein levels (Figure 4H). Last but not least, ChIP results showed that shTLR4 significantly decreased H3K27me3 levels in both SLC26A2 and ST6GalNAc6 promoter regions (Figure 4I). Taking these results together, TLR4 may regulate NF-κB /BGN loop for epigenetic silencing of SLC26A2 and ST6GalNAc6.

## 4. Discussion

In this study, we showed that BGN may mediate TLR4/NF-κB epigenetic silencing of SLC26A2 and ST6GalNAc6 through EZH2-containing PRC2. Compared to HIEC6 cells (healthy intestinal epithelial cells), DLD-1, HT29, and LS174T cells having low expression levels of ST6GalNAc6 and SLC26A2 mRNA were selected for experiments [12]. Previously, we showed that immunosuppressive Siglec ligands, disialyl Lewis^a^, and sialyl 6-sulfo Lewis^x^ glycans, are preferentially expressed in non-malignant colonic epithelial cells and tend to disappear in colon cancer cells [6,7]. We proposed that this is due to NF-κB-mediated PRC2 epigenetic silencing of SLC26A2 and ST6GalNAc6 during the very early stages of colon cancer [12].

Siglecs play dual roles as pro-oncogenic or anti-oncogenic molecules during cancer progression. The double features of Siglecs may reflect the dual functions of macrophages, that is, that M1 macrophage polarization supports tumor cell clearance and M2 macrophage polarization enhances cancer progression [28]. *Fusobacterium nucleatum* promotes M2 macrophage polarization via a TLR4-dependent mechanism in the colorectal tumor [29]. In our previous findings, we confirmed that the interaction of Siglecs with their glycan ligands suppresses macrophages and inhibits production of pro-oncogenic inflammation mediators such as COX2 [9] and proposed that the loss of Siglec binding glycan ligands will facilitate colorectal cancer carcinogenesis. It is worth noting that healthy epithelial cells express a considerable amount of immunosuppressive Siglec ligands, which are lost during the carcinogenesis process.

TLRs are the crucial molecules involved in inflammation-driven carcinogenesis. Expression of TLR2 or TLR4 mRNA is higher in patients’ tumor samples, which was suggested to be a marker in human colorectal cancer [30,31]. However, the expression level of TLR2 or TLR4 is only slightly increased in colon cancer samples according to the GENT2 database. Instead of TLR2 or TLR4, BGN (which is significantly increased in colon cancer) may be a more useful marker in human colorectal cancer [32].

Experimental evidence shows that gut microbiota may serve as prognostic markers and be involved in colorectal cancer initiation and progression [33]. Some colonic microbiota, such as *Fusobacterium nucleatum* and *Bacteroides fragilis*, are especially potent in activating NF-κB activity [34,35]. *Fusobacterium nucleatum* is significantly elevated in human colorectal tumors compared to that in adjacent healthy tissue [36], *Fusobacterium nucleatum* stimulates NF-κB by activating TLR4 signaling and facilitates tumor development [31,37]. Whether colonic microbiota, such as *Fusobacterium nucleatum*, play roles in the suppression of Siglec ligand expression in colonic epithelial cells will require further investigation.

BGN is involved in various aspects of cancer biology, including cancer cell proliferation, invasion [38], epithelial–mesenchymal transition [39], angiogenesis [40], chemotherapy resistance [23], and patient prognosis [41], but the role of BGN during the early stages of cancer has not yet been elucidated. BGN expression is increased in many cancers including bladder cancer [39], colorectal cancer [42], gastric cancer [43], esophageal cancer [44], prostate cancer [40], and endometrial cancer [45] and various factors modulate its expression, including p38 [46], HIF-1 [47], TGF-β/Smad4 [48], and NF-κB [26]. NF-κB may serve a crucial factor at the early stage of colorectal cancer progression, and we demonstrated that it may play a crucial role in the regulation of BGN gene expression. The expression of BGN mRNA was dramatically increased in stage I colorectal cancer samples in the TCGA database. This underscores the role of BGN during colon carcinogenesis. BGN binds to both TLR2 and TLR4, but BGN triggers TLR signaling mainly through TLR4 in a MyD88-dependent manner and causes activation of the NF-κB pathway [49]. Our data also suggest that the introduction of shRNA for BGN as well as TLR4 but not TLR2 significantly restore immunosuppressive Siglec ligand expression. However, in TLR2-dominant cells such as DLD-1, TLR2 seems to play a minor role in suppressing expression of immunosuppressive glycan ligands. The mechanism must be investigated further. 

The most commonly accepted feature in NF-κB dynamics is the presence of oscillations upon external stimuli [50,51]. NFKBIA and TNFAIP3 are two major feedback loops of NF-κB [52,53]. Cancer cells are heterogeneous. Sakamoto et al. reported that NF-κB activation was observed in 40% (35 of 88) of colorectal carcinomas, and NF-κB activation was determined by p65 nuclear staining, which was observed in >50% of the cancer cells in the carcinoma tissues [24]. Colorectal cancer patient samples (*n* = 626) display much lower NFKBIA (0.643-fold) and TNFAIP3 (0.745-fold) gene expression compared to healthy control (*n* = 51) based on the TCGA database (http://firebrowse.org/) (Appendix A). The lower expression of NFKBIA and TNFAIP3 may explain why a part of the patients displayed much stronger NF-κB activity.

## Figures and Tables

**Figure 1 cells-09-00397-f001:**
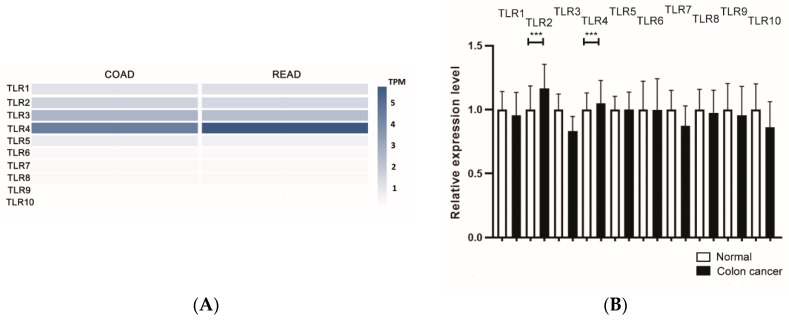
Toll-like receptor 4 (TLR4) mediates the suppression of immunosuppressive Siglec-7 glycan ligands in colorectal cancers. (**A**) Comparison of TLR1–TLR10 amount expression in COAD (Colon adenocarcinoma) and READ (Rectum adenocarcinoma) by Gene Expression Profiling Interactive Analysis (GEPIA2). Gene expression level was calculated by transcript per million (TPM). (**B**) TLR1–TLR10 expression level in normal (*n* = 397) and colon cancer samples (*n* = 3775) from the GENT2 database. *** *p* < 0.001. Results are shown as the relative fold increase compared with normal colon control. (**C**) Gene expression analysis of TLR4 in human colon cancer cells (Ca) and non-malignant colonic epithelial cells (N) prepared from same patients (*n* = 10). ** *p* < 0.01. (**D**) Gene expression analysis of TLR2 or TLR4 in normal intestinal cells (HIEC6), DLD-1, HT29, and LS174T cells. (**E**) Gene expression analysis of TLR2 in DLD1 and LS174T after the introduction of shRNAs for TLR2. (**F**) Gene expression analysis of TLR4 in HT-29 and LS174T after the administration of shRNAs for TLR4. (**G**) SLC26A2 expression in colon cancer cells after the administration of shRNAs for TLR2 and TLR4. (**H**) ST6GalNAc6 expression in colon cancer cells after the administration of shRNAs for TLR2 and TLR4. (**I**) Flow cytometry analysis of expression of disialyl Lewis^a^ (top panels) and sialyl 6- sulfo Lewis^x^ (bottom panels) on DLD-1 and LS174T cells, respectively, after the administration of shRNAs for TLR2. (**J**) Fluorescence intensity results of flow cytometry analysis in (**I**) are shown. The relative expression levels of disialyl Lewis^a^ stained with FH7 antibody or sialyl 6-sulfo Lewis^x^ stained with G72 antibody are shown as mean fluorescence intensity (MFI) compared with the staining of control IgG or IgM. *** *p* < 0.001. (**K**) Flow cytometry analysis of expression of disialyl Lewis^a^ (top panels, FH7) or sialyl 6- sulfo Lewis^x^ (bottom panels, G72) on HT-29 or LS174T cells, respectively, after the administration of shRNAs for TLR4. (**L**) Fluorescence intensity results of flow cytometry analysis in (**K**) are shown. The relative expression levels of disialyl Lewis^a^ and sialyl 6-sulfo Lewis^x^ are shown as mean fluorescence intensity (MFI) compared with the staining of control IgG or IgM. *** *p* < 0.001.

**Figure 2 cells-09-00397-f002:**
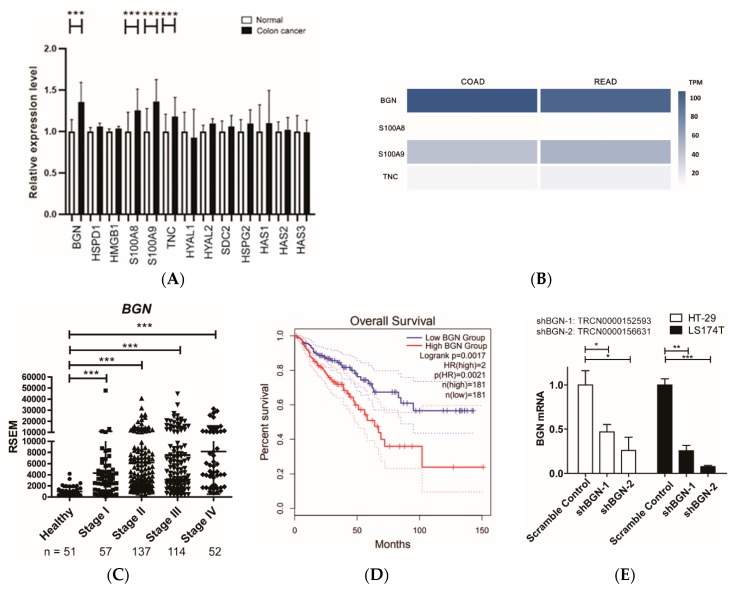
Identification of BGN as a potential TLR4 ligand involved in suppression of Siglec ligands in the early stage of colon cancer carcinogenesis. (**A**) Gene expression analysis of TLR4 ligands—BGN, HSPD1, HMGB1, S100A8, S100A9, TNC, heparan sulfate proteoglycan active enzymes (SDC2 or HSPG2), hyaluronidases (HYAL1 and HYAL2) and hyaluronan synthase (HAS1, HAS2, and HAS3)—in normal (*n* = 397) and colon cancer samples (*n* = 3775) from the GENT2 database. ***, *p* < 0.001. Results are shown as the relative fold increase compared with normal colon control. (**B**) Comparison of BGN, S100A8, S100A9, and TNC amount expression in COAD and READ by GEPIA2. Gene expression level was calculated by transcript per million (TPM). (**C**) Large-scale RNA-Seq transcriptome analysis of BGN in healthy colorectal tissues and tissues from four stages of colorectal cancer from the TCGA database. n, number of patients; *** *p* < 0.001. (**D**) BGN was inversely correlated with the overall survival of patients with colorectal cancer. Overall survival analyses were performed using the GEPIA2 online platform. The solid line shows the survival curve, and the dotted line shows the 95% confidence interval. Red lines indicate patients with expression above the median, and blue lines indicate patients with expression below the median. Log-rank *p* < 0.05 was considered statistically significant. HR, hazard ratio. (**E**) BGN expression in HT-29 and LS174T after the administration of shRNAs for BGN. (**F**) Flow cytometry analysis of expression of disialyl Lewis^a^ on HT-29 cells (top panels) and sialyl 6- sulfo Lewis^x^ on LS174T cells (bottom panels), after the administration of shRNAs for BGN. (**G**) Fluorescence intensity results of flow cytometry analysis in (**H**) are shown. The relative expression levels of disialyl Lewis^a^ and sialyl 6-sulfo Lewis^x^ are shown as mean fluorescence intensity (MFI) compared with the staining of control IgG or IgM. *** *p* < 0.001. (**H**) Restoration of Siglec-7 binding by introduction of shRNAs for BGN in LS174T colon cancer cells. The relative levels of Siglec-7 are shown as the relative mean fluorescence intensity (MFI) compared with the control group IgG. *** *p* < 0.001.

**Figure 3 cells-09-00397-f003:**
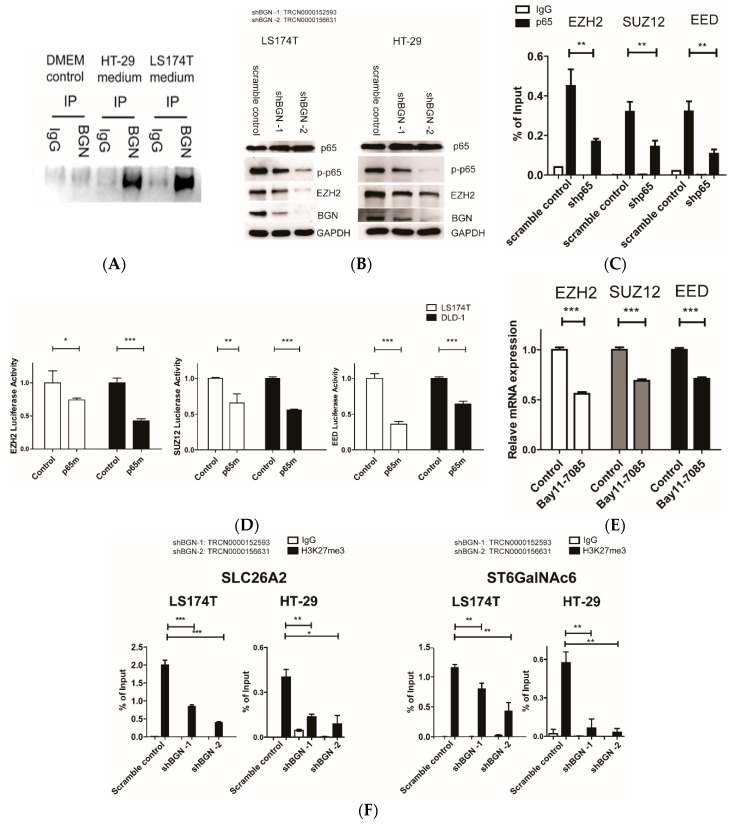
Suppression of BGN inactivates NF-κB and decreases PRC2 activity in the promoter regions of SLC26A2 and ST6GalNAc6. (**A**) Western blot experiments of BGN in HT-29 and LS174T cultured medium. DMEM medium serves as a negative control. (**B**) Western blot experiments of phosphorylated p65 (p-p65) in LS174T and HT-29 after the introduction of shRNA for BGN. (**C**) ChIP/RT-qPCR results of p65 levels in EZH2, SUZ12, and EED regulatory regions in DLD-1 cells. (**D**) Mutational analysis of the p65-binding site in EZH2 p(+994/+841 in pGL3 promoter vector), SUZ12 p(−231/+69 in pGL3 basic vector), and EED p(−348/−49 in pGL3 basic vector). The mutated p65 construct (p65m) had a mutation of p65 binding site at bases +885 to +876 (from CCCCTAAAGC to TTTTTTTTT) in EZH2 and at bases −114 to −105 (from GGGGAATCCGC to AAAAAATAAAA) in SUZ12 and a mutation of p65 binding site at bases −219 to −210 (from GGGTACTTTCCC to AATACAAAAA) in EED. The control and mutated p65 constructs were transiently expressed in DLD-1 and LS174T cells for reporter assays. ***, *p* < 0.001;**, *p* < 0.01;*, *p* < 0.05 (*n* = 3 with mean ± SD shown). (**E**) EZH2, SUZ12, and EED mRNA in LS174T cells after administration with Bay11-7085 (10 μg/ml) for 6 h were analyzed by RT-qPCR. (**F**) ChIP/RT-qPCR results of H3K27me3 level in SLC26A2 and ST6GalNAc6 promoter regions in LS174T and HT-29 cells.

**Figure 4 cells-09-00397-f004:**
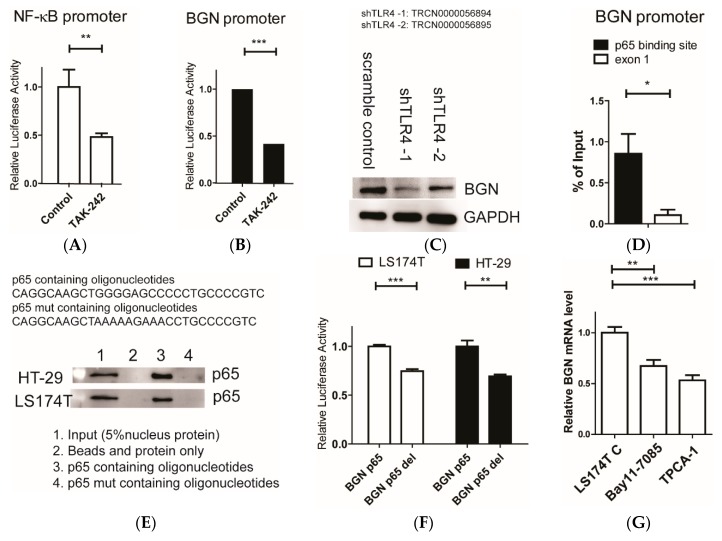
TLR4 modulates BGN activity through NF-κB. (**A**,**B**) Promoter analysis of NF-κB (pGL4.31) and BGN promoter (−823/−1) after treatment with TLR4 inhibitor (TAK-242) for 1 d. Firefly luciferase activity was normalized to Renilla luciferase activity. ***, *p* < 0.001;**, *p* < 0.01 (*n* = 3 with mean ± SD shown). (**C**) Western blot experiments of BGN in LS174T after the administration of shRNA for TLR4. (**D**) ChIP/RT-qPCR results of p65 levels in BGN promoter regions in LS174T cells. (**E**) DAPA of the p65-binding site in the BGN (−656 to −627, CAGGCAAGCTGGGGAGCCCCCTGCCCCGTC) promoter. p65 binding to biotinylated oligonucleotides containing either wild-type (p65) or a mutated p65-binding (p65 mut) sequence was assayed by DNA affinity precipitation and analyzed by western blotting using nuclear extracts of HT-29 and LS174T cells. (**F**) Promoter analysis BGN promoter (−823/−1) and BGN p65 deletion promoter after transfection for 1 d. (**G**) BGN mRNA in LS174T cell after administration with Bay11-7085 (10 μg/ml) or TPCA-1 (5 μg/ml) for 6 h analyzed by RT-qPCR. (**H**) Western blot experiments of BGN in LS174T after administration with Bay11-7085 (10 μg/ml) or TPCA-1 (5 μg/ml) for 3 d. (**I**) ChIP/RT-qPCR results of H3K27me3 level in SLC26A2 and ST6GalNAc6 promoter regions in LS174T cells.

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
