# Peer review of "BGN/TLR4/NF-κB Mediates Epigenetic Silencing of Immunosuppressive Siglec Ligands in Colon Cancer Cells"

_cells, 2020, doi:10.3390/cells9020397_

Round 1

Reviewer 1 Report

In this paper, the authors reveal a novel axis involving the activation of NF-kappaB and leading to the downregulation of the immunosuppressive ligands and the appearance of a couple of cancer-associated glycans. Such axis is mediated by the TLR4 activation by BGN, which leads itself to a positive feedback loop mediated by an NF-kappaB- dependent BGN expression.

In my opinion this in an interesting study, the results are well described and many are replicated in different tumor cell lines of colon cancer. However, some issues shall be addressed before making it acceptable for publication. 

1.- I think that the authors prove that the axis pointed out is present in tumor cell lines, but in my opinion they do not present enough evidence to show that this is the case in “early stage colon cancers”. From Figure 2C it is clear that this is the case in all the stages of the disease. Furthermore, they are using cancer cell lines so I’d change that part of the title to “in colon cancer cells”.

2.- I find always puzzling in this work, as in others, NF-kappaB activation in cancer cell lines seems to be “constant” and “high”. However the activity of NF-kappaB is known to be regulated by negative feedbacks (PMID: 12424381 ) and at single cell level this leads to a mixture of cells activated and inactivated “oscillating” in an inflammatoroy microenvironment (e.g. PMID: 20581820, PMID: 29673148). I’d be glad to read in the discussion how do they place their observations within this more general single-cell context: how do cells overcome this regulation? is NF-kappaB always nuclear? Just in a fraction of cells? In all of them? 

3.- The evidence provided, in my view, points out to BGN as a potential but maybe not the only driver of the NF-kappaB activation observed. Since the authors observe this phenomenon in cancer cell lines it means that the cells in culture release BGN, which shall be detectable in the supernatant. To show that it is there would be important. Furthermore, to quantify its amount against other cytokines involved in NF-kappaB activation (e.g. il1beta, TNFalpha) would certainly be a plus.  

Some minor comments

A graphical summary with the model proposed would be helpful for a reader.  In plots of fig 1 G and H and similar throughout the paper it would be helpful to see to which gene they refer to. 

Author Response

Reviewer 1

1.- I think that the authors prove that the axis pointed out is present in tumor cell lines, but in my opinion they do not present enough evidence to show that this is the case in “early stage colon cancers”. From Figure 2C it is clear that this is the case in all the stages of the disease. Furthermore, they are using cancer cell lines so I’d change that part of the title to “in colon cancer cells”.

Thank you. We don’t have enough evidence support those events happened in the early stage of cancer carcinogenesis. And we performed those experiments using only colon cancer cells. So we changed the title to “BGN/TLR4/NF-kB-mediates epigenetic silencing of immunosuppressive siglec ligands “in colon cancer cells,” as suggested by the reviewer.

2.- I find always puzzling in this work, as in others, NF-kappaB activation in cancer cell lines seems to be “constant” and “high”. However the activity of NF-kappaB is known to be regulated by negative feedbacks (PMID: 12424381 ) and at single cell level this leads to a mixture of cells activated and inactivated “oscillating” in an inflammatoroy microenvironment (e.g. PMID: 20581820, PMID: 29673148).

I’d be glad to read in the discussion how do they place their observations within this more general single-cell context: how do cells overcome this regulation? is NF-kappaB always nuclear? Just in a fraction of cells? In all of them?

The most commonly accepted feature in NF-kB dynamics is the presence of oscillations upon external stimuli [50,51]. NFKBIA and TNFAIP3 are two major feedback loops of NF-kB [52,53]. Cancer cells are heterogeneous. Kei S. reported that NF-κB activation was observed in 40% (35 of 88) of colorectal carcinomas and NF-kB activation was determined by p65 nuclear staining, which was observed in >50% of the cancer cells in the carcinoma tissues [24]. Colorectal cancer patient samples (n = 626) display much lower NFKBIA (0.643 folds) and TNFAIP3 (0.745 folds) gene expression compared to healthy control (n= 51) based on TCGA database (http://firebrowse.org/) (Figure S5). The lower expression of NFKBIA and TNFAIP3 may explain why a part of patients display much stronger NF-kB activity.

3.- The evidence provided, in my view, points out to BGN as a potential but maybe not the only driver of the NF-kappaB activation observed. Since the authors observe this phenomenon in cancer cell lines it means that the cells in culture release BGN, which shall be detectable in the supernatant. To show that it is there would be important. Furthermore, to quantify its amount against other cytokines involved in NF-kappaB activation (e.g. il1beta, TNFalpha) would certainly be a plus.

According to the comments of the reviewer, we performed new experiments to detect BGN in HT-29 and LS174T cultured medium by immunoprecipitation-western blotting method.

Below is the newly added Figure 3A. (see attached pdf file)

Reviewer 2 Report

The study by Huang and colleagues proposes shows TLR4 signaling to be crucial for downregulation of disialyl Lewisa and sialyl 6-sulfo Lewisx glycans in human colon cancer cell lines, which serve as ligands for the immune suppressive molecule Siglec-7. In particular, the authors propose a  BGN-TLR4-NF-kB-BGN loop resulting in epigenetic silencing of SLC26A2 and ST6GalNAc6 to account for the observed downregulation of disialyl Lewisa and sialyl 6-sulfo Lewisx glycans. Although the findings are highly interesting, some major comments have to be addressed prior to considering the manuscript for publication.

 Major comments:

Figure 1B. Although being significant, the changes in relative expression levels for TLR2 and TLR4 between normal colon and colon cancer are only minor. The authors should include own data of primary tumors and/or at least protein data here to substantiate their initial hypothesis of TLR4 being highly increased in colon cancer. Figure 1C-G. Although the knockdown experiments nicely depict the increase the Siglec-7 glycan ligands in the colon cancer cell lines upon application of shTLR4, the authors should include some missing experiment here. The differential effects of shTLR4 and shTLR2 are only completely shown in one single cell line (LS174T) and data for shTLR2 in HT29 and shTLR4 in DLD1 should be included in Figures 1E-H for mRNA and protein levels of disialyl Lewisa and sialyl 6-sulfo Lewisx.  In addition, shTLR2 seems to have a minor effect on disialyl Lewisa  in DLD1 cells (Fig. 1G), the authors should provide the quantities of the respective populations for all flow cytometric analysis and discuss this minor effect of shTLR2.

Figure 2F. The authors propose BGN as ligand for TLR4 in the colon cancer cell lines and show shBGN to restore normal glycan expression. It is unclear to the referee, which cell line exactly was applied for the anaylsis of disialyl Lewisa and sialyl 6-sulfo Lewisx. Again, the authors should show the presented effects of shBGN in three independent cell lines (as for requested for Fig. 1, see above).

Figure 2/ Figure 3. The authors should show that exposure to BGN is really resulting in a TLR4-NF-kB response as suggested to validate the hypothesis of BGN as ligand for TLR4 in an NF-kB-dependent manner in the applied colon cancer cell lines.

Figure 3E. As stated by the authors, one major hypothesis is that “BGN may serve as a trigger to initiate NF-232 κB/PRC2-mediated epigenetic silencing of SLC26A2 and ST6GalNAc6” (lines 232-233). To validate this key finding, the results depicted in Fig. 3E should be shown in at least one additional cell line.

Figure 3/4. The presented data nicely assess the effects of knockdowns but lack effects of BGN as a ligand for the proposed BGN-TLR4-NF-kB-BGN loop resulting in epigenetic silencing of SLC26A2 and ST6GalNAc6 (see also issue 4). These data should be included in the manuscript.

 Minor comments:

Figure 3A. Western blot analysis against p65 should be included to allow comparison to p-p65.

Figure 3C. It is unclear in the text which reporter gene activity was assessed exactly, please clarify in the text and figure labeling.

Labeling of axis should be improved to better guide the reader (eg Fig. 1B; Fig. 1G,H, Fig.2F).

Author Response

Reviewer 2 (please see the attached pdf file for figured items)

Major comments:

Figure 1B. Although being significant, the changes in relative expression levels for TLR2 and TLR4 between normal colon and colon cancer are only minor. The authors should include own data of primary tumors and/or at least protein data here to substantiate their initial hypothesis of TLR4 being highly increased in colon cancer.

According to the comments of the reviewer, we analyzed TLR4 mRNA in human colon cancer (Ca) tissues and nonmalignant mucosa (N) prepared from same patient (n = 10) and added this result in Figure 1L.

Figure 1C-G. Although the knockdown experiments nicely depict the increase the Siglec-7 glycan ligands in the colon cancer cell lines upon application of shTLR4, the authors should include some missing experiment here. The differential effects of shTLR4 and shTLR2 are only completely shown in one single cell line (LS174T) and data for shTLR2 in HT29 and shTLR4 in DLD1 should be included in Figures 1E-H for mRNA and protein levels of disialyl Lewisa and sialyl 6-sulfo Lewisx.

During these experiments, we carefully checked mRNA levels of TLR2 and TLR4 in DLD1, HT29 and LS174T. Compare to HIEC6 (normal intestinal epithelial cells), DLD-1 and LS174T cells having high expression levels of TLR2 were selected in the study of TLR2 knocking down experiments. Similarly, HT-29 and LS174T having high expression levels of TLR4 were selected in the study of TLR4 knocking down experiments

We added the results on mRNA levels in DLD1, HT29 and HIEC6 cells in Figure 1C of the revised manuscript.

Indeed, in Figure 1F-G, we show shTLR4 can induce both ST6GalNAc6 and SLC26A2 expression in HT-29 and LS174T cells. Disialyl Lewisa is synthesized on the type 1 chain glycans through ST6GAlNAc6, and sialyl 6-sulfo Lewisx is synthesized on the type 2 chain glycans with the aid of SLC26A2. Among cultured colon cancer cells, there is a type 1 chain glycan-dominant group of cells, and a type 2 chain glycan-dominant group of cells. For the induction of disialyl Lewisa glycan and ST6GAlNAc6 mRNA, the HT-29 belonging to the former group is more appropriate, and for observation of sialyl 6-sulfo Lewisx glycan expression and DTDST mRNA, the LS174T cells are more suitable. This is the main reason why we use different set of the cell line in the experiments.

In addition, shTLR2 seems to have a minor effect on disialyl Lewisa in DLD1 cells (Fig. 1G), the authors should provide the quantities of the respective populations for all flow cytometric analysis and discuss this minor effect of shTLR2.

According to the comments of the reviewer, we added the quantitative results in all flow cytometric analysis results in the manuscript as follows.

Figure 1I             Figure 1K             Figure 2G

According to the comments of the reviewer, we added the description in page 12 lane 361. “However, in TLR2 dominant cells such as DLD-1, TLR2 seems to play a minor role in suppressing expression of immune-suppressive glycan ligand. The mechanism needs to be investigated further”.

Figure 2F. The authors propose BGN as ligand for TLR4 in the colon cancer cell lines and show shBGN to restore normal glycan expression. It is unclear to the referee, which cell line exactly was applied for the anaylsis of disialyl Lewisa and sialyl 6-sulfo Lewisx. Again, the authors should show the presented effects of shBGN in three independent cell lines (as for requested for Fig. 1, see above).

Thank you. we clarified figure labeling in Figure 2F to indicate the names of the cells used.

Figure 2/ Figure 3. The authors should show that exposure to BGN is really resulting in a TLR4-NF-kB response as suggested to validate the hypothesis of BGN as ligand for TLR4 in an NF-kB-dependent manner in the applied colon cancer cell lines.

Thank you. We cited the paper which describes BGN as ligand for TLR4 in an NF-kB-dependent manner in HT-29 cell line in Page 8, line 245 ff..

BGN is a small leucine rich proteoglycan ECM protein. Soluble biglycan directly interacts with TLR-2 and TLR-4 [21]. Mechanistically, BGN is known to rapidly activate NF-kB by engaging TLR2 and TLR4 [22]. In HT-29, suppression of BGN can decrease NF-kB activity [23]. To access BGN could secrete to cultured medium and serve as a potential ligand to TLR2/4 in colon cancer cells, we observed BGN in HT-29 and LS174T cultured medium by immunoprecipitation-western blotting method (Figure 3A).

Figure 3E. As stated by the authors, one major hypothesis is that “BGN may serve as a trigger to initiate NF-κB/PRC2-mediated epigenetic silencing of SLC26A2 and ST6GalNAc6” (lines 232-233). To validate this key finding, the results depicted in Fig. 3E should be shown in at least one additional cell line.

Thank you. According to the comments of the reviewer, we added the ChIP results of HT-29 in Figure 3E.

Figure 3/4. The presented data nicely assess the effects of knockdowns but lack effects of BGN as a ligand for the proposed BGN-TLR4-NF-kB-BGN loop resulting in epigenetic silencing of SLC26A2 and ST6GalNAc6 (see also issue 4). These data should be included in the manuscript.

Thank you. We cited the paper to show BGN is a ligand for TLR4 and performed new experiments to demonstrate BGN is present in culture medium. To verify BGN is secreted to cultured medium and serves as a potential ligand to TLR2/4 in colon cancer cells, we observed BGN in HT-29 and LS174T cultured medium by immunoprecipitation-western blotting method (Figure 3A).

Minor comments:

Figure 3A. Western blot analysis against p65 should be included to allow comparison to p-p65.

Thank you. According to the comments of the reviewer, we added the p65 results of western blot in Figure 3A (now is Figure 3B).

Figure 3C. It is unclear in the text which reporter gene activity was assessed exactly, please clarify in the text and figure labeling.

Thank you. According to the comments of the reviewer, we clarified figure labeling in Figure 3C (now is Figure 3D).

Labeling of axis should be improved to better guide the reader (eg Fig. 1B; Fig. 1G,H, Fig.2F).

Thank you. We improved the labeling of axis in Figure 1B; Figure 1G; Figure 1H; and Figure 2F.

1G                   1H                     2F

Round 2

Reviewer 2 Report

Major comments:

1. Figure 1B. Although being significant, the changes in relative expression levels for TLR2 and TLR4 between normal colon and colon cancer are only minor. The authors should include own data of primary tumors and/or at least protein data here to substantiate their initial hypothesis of TLR4 being highly increased in colon cancer.

According to the comments of the reviewer, we analyzed TLR4 mRNA in human colon cancer (Ca) tissues and nonmalignant mucosa (N) prepared from same patient (n =10) and added this result in Figure 1L.

Response to revised version:  The newly added data properly address the issue. However, it is unclear why the author chose to place the new panel as Figure 1L and not as Fig. 1C directly after the data from GENT2 database. The figure design should be revised to better guide the reader throughout the manuscript. In addition, the paper lacks an ethical statement regarding the human samples, which must be included in the revised manuscript.

2. Figure 1C-G. Although the knockdown experiments nicely depict the increase the Siglec-7 glycan ligands in the colon cancer cell lines upon application of shTLR4, the authors should include some missing experiment here. The differential effects of shTLR4 and shTLR2 are only completely shown in one single cell line (LS174T) and data for shTLR2 in HT29 and shTLR4 in DLD1 should be included in Figures 1E-H for mRNA and protein levels of disialyl Lewisa and sialyl 6-sulfo Lewisx.

During these experiments, we carefully checked mRNA levels of TLR2 and TLR4 in DLD1, HT29 and LS174T. Compare to HIEC6 (normal intestinal epithelial cells), DLD-1 and LS174T cells having high expression levels of TLR2 were selected in the study of TLR2 knocking down experiments. Similarly, HT-29 and LS174T having high expression levels of TLR4 were selected in the study of TLR4 knocking down experiments

We added the results on mRNA levels in DLD1, HT29 and HIEC6 cells in Figure 1C of the revised manuscript.

Indeed, in Figure 1F-G, we show shTLR4 can induce both ST6GalNAc6 and SLC26A2 expression in HT-29 and LS174T cells. Disialyl Lewisa is synthesized on the type 1 chain glycans through ST6GAlNAc6, and sialyl 6-sulfo Lewisx is synthesized on the type 2 chain glycans with the aid of SLC26A2. Among cultured colon cancer cells, there is a type 1 chain glycan-dominant group of cells, and a type 2 chain glycan- dominant group of cells.                 For the induction of disialyl Lewisa glycan and ST6GAlNAc6 mRNA, the HT-29 belonging to the former group is more appropriate, and for observation of sialyl 6-sulfo Lewisx glycan expression and DTDST mRNA, the LS174T cells are more suitable.   This is the main reason why we use different set of the cell line in the experiments.

In addition, shTLR2 seems to have a minor effect on disialyl Lewisa in DLD1 cells (Fig. 1G), the authors should provide the quantities of the respective populations for all flow cytometric analysis and discuss this minor effect of shTLR2.

According to the comments of the reviewer, we added the quantitative results in all flow cytometric analysis results in the manuscript as follows.

According to the comments of the reviewer, we added the description in page 12 lane 361.“However, in TLR2 dominant cells such as DLD-1, TLR2 seems to play a minor role in suppressing expression of immune-suppressive glycan ligand. The mechanism needs to be investigated further”.

Response to revised version:  The issues are properly addressed by the added data. The authors may additionally include the reasoning regarding the used cell lines (see above) in the discussion section.

3. Figure 2F. The authors propose BGN as ligand for TLR4 in the colon cancer cell lines and show shBGN to restore normal glycan expression. It is unclear to the referee, which cell line exactly was applied for the anaylsis of disialyl Lewisa and sialyl 6-sulfo Lewisx. Again, the authors should show the presented effects of shBGN in three independent cell lines (as for requested for Fig. 1, see above).

Thank you. we clarified figure labeling in Figure 2F to indicate the names of the cells used.

Response to revised version:  The issue is properly addressed by the added data and the discussion of major comment 2.

4. Figure 2/ Figure 3. The authors should show that exposure to BGN is really resulting in a TLR4-NF-kB response as suggested to validate the hypothesis of BGN as ligand for TLR4 in an NF-kB-dependent manner in the applied colon cancer cell lines.

Thank you. We cited the paper which describes BGN as ligand for TLR4 in an NF-kB- dependent manner in HT-29 cell line in Page 8, line 245 ff..

BGN is a small leucine rich proteoglycan ECM protein. Soluble biglycan directly interacts with TLR-2 and TLR-4 [21]. Mechanistically, BGN is known to rapidly activate NF-kB by engaging TLR2 and TLR4 [22]. In HT-29, suppression of BGN can decrease NF-kB activity [23]. To access BGN could secrete to cultured medium and serve as a potential ligand to TLR2/4 in colon cancer cells, we observed BGN in HT-29 and LS174T cultured medium by immunoprecipitation-western blotting method (Figure 3A).

Response to revised version:  The issue is properly addressed by the added data.

5. Figure 3E. As stated by the authors, one major hypothesis is that “BGN may serve as a trigger to initiate NF-κB/PRC2-mediated epigenetic silencing of SLC26A2 and ST6GalNAc6” (lines 232-233). To validate this key finding, the results depicted in Fig. 3E should be shown in at least one additional cell line.

Thank you. According to the comments of the reviewer, we added the ChIP results of HT-29 in Figure 3E.

Response to revised version:  The issue is properly addressed by the added data.

6. Figure 3/4. The presented data nicely assess the effects of knockdowns but lack effects of BGN as a ligand for the proposed BGN-TLR4-NF-kB-BGN loop resulting in epigenetic silencing of SLC26A2 and ST6GalNAc6 (see also issue 4). These data should be included in the manuscript.

Thank you. We cited the paper to show BGN is a ligand for TLR4 and performed new experiments to demonstrate BGN is present in culture medium. To verify BGN is secreted to cultured medium and serves as a potential ligand to TLR2/4 in colon cancer cells, we observed BGN in HT-29 and LS174T cultured medium by immunoprecipitation-western blotting method (Figure 3A).

Response to revised version:  The issue is properly addressed by the added data.

Minor comments:

Figure 3A. Western blot analysis against p65 should be included to allow comparison to p-p65.

Thank you. According to the comments of the reviewer, we added the p65 results of western blot in Figure 3A (now is Figure 3B).

Figure 3C. It is unclear in the text which reporter gene activity was assessed exactly, please clarify in the text and figure labeling.

Thank you. According to the comments of the reviewer, we clarified figure labeling in Figure 3C (now is Figure 3D).

Labeling of axis should be improved to better guide the reader (eg Fig. 1B; Fig. 1G,H, Fig.2F).

Thank you. We improved the labeling of axis in Figure 1B; Figure 1G; Figure 1H; and Figure 2F.

Response to revised version:  All minor comments are adequately addressed.

Author Response

Response to Reviewer 2 Comments

Major comments:

Point 1: Response to revised version: The newly added data properly address the issue. However, it is unclear why the author chose to place the new panel as Figure 1L and not as Fig. 1C directly after the data from GENT2 database. The figure design should be revised to better guide the reader throughout the manuscript. In addition, the paper lacks an ethical statement regarding the human samples, which must be included in the revised manuscript.

Response 1: Thank you. According to the comments of the reviewer, we place Figure 1L into Figure 1C directly after the data from GENT2 database. We also add the ethical statement in Page 2 line 68.

Point 2: Response to revised version: The issues are properly addressed by the added data. The authors may additionally include the reasoning regarding the used cell lines (see above) in the discussion section.

Response 2: Thank you. According to the comments of the reviewer, we added the reasoning regarding the used cell lines in description in Page 12 line 317 as follows:.

Compared to HIEC6 cells (healthy intestinal epithelial cells), DLD-1, HT29, and LS174T cells having low expression levels of ST6GalNAc6 and SLC26A2 mRNA were selected for experiments.
